# Epitranscriptomics in Normal and Malignant Hematopoiesis

**DOI:** 10.3390/ijms21186578

**Published:** 2020-09-09

**Authors:** Crescenzio Francesco Minervini, Elisa Parciante, Luciana Impera, Luisa Anelli, Antonella Zagaria, Giorgina Specchia, Pellegrino Musto, Francesco Albano

**Affiliations:** 1Department of Emergency and Organ Transplantation (D.E.T.O.)-Hematology and Stem Cell Transplantation Unit-University of Bari “Aldo Moro”, 70124 Bari, Italy; eziominervini@gmail.com (C.F.M.); elisaparciante@libero.it (E.P.); lucianaim@hotmail.com (L.I.); luisa.anelli@uniba.it (L.A.); antonellazagaria@hotmail.com (A.Z.); pellegrino.musto@uniba.it (P.M.); 2Former Full Professor of Hematology–University of Bari “Aldo Moro”, 70124 Bari, Italy; specchiagiorgina@gmail.com

**Keywords:** epitranscriptomics, hematopoiesis, hematological malignancies

## Abstract

Epitranscriptomics analyze the biochemical modifications borne by RNA and their downstream influence. From this point of view, epitranscriptomics represent a new layer for the control of genetic information and can affect a variety of molecular processes including the cell cycle and the differentiation. In physiological conditions, hematopoiesis is a tightly regulated process that produces differentiated blood cells starting from hematopoietic stem cells. Alteration of this process can occur at different levels in the pathway that leads from the genetic information to the phenotypic manifestation producing malignant hematopoiesis. This review focuses on the role of epitranscriptomic events that are known to be implicated in normal and malignant hematopoiesis, opening a new pathophysiological and therapeutic scenario. Moreover, an evolutionary vision of this mechanism will be provided.

## 1. Introduction

Recent technological advances have made it possible to gain evidence of a wide range of post-transcriptional RNA modifications that play pivotal roles in fundamental biological processes including the regulation of gene expression. The information implicated in this field of study is collectively known as epitranscriptomics.

During maturation, RNA becomes the substrate of various enzymes that can introduce chemical modifications into ribonucleotide residues. Usually, chemical alterations may occur in the base, or at the 2′-hydroxyl of the ribose, or both, and sometimes multiple sequential changes may involve the same nucleoside [1].

RNA modifications can vary significantly in terms of the type of modification, location, abundance, and distribution both between RNA molecules and between organisms and organelles. Environmental and growth conditions can also affect the pattern of change [2].

The majority of modified nucleosides are present in transfer and ribosomal RNAs (tRNA, rRNA) but they can also occur in small non-coding RNAs such as spliceosomal RNAs (snRNAs), small nucleolar RNAs (sno RNAs), and regulatory RNAs such as siRNAs, miRNAs, and piRNAs as well as in long non-coding RNA (lncRNA) and messenger RNA (mRNA). According to the MODOMICS database (https://iimcb.genesilico.pl/modomics/), ~170 different RNA modifications have been identified in coding and non-coding RNAs [3,4,5,6,7].

In eukaryotes, the most common reversible modification is N6-methyladenosine (m6A), which occurs at the RRm6AACH motif in mRNA [8,9,10]. Alternative methylations involve the carbon in the fifth position on cytosine (5-methylcytidine/m5C) and the nitrogen in the first position on adenine (N1-methyladenosine/m1A). Other well-characterized modifications include pseudouridylation (Ψ) and RNA editing [11,12,13].

Pseudouridine is the C5-glycoside isomer of uridine and is the most abundant internal RNA modification in stable non-coding RNAs [14]. Ψ is necessary to guarantee the correct secondary/tertiary structure of rRNA and tRNA, directly affecting translational events. Moreover, in snRNA, it can affect snRNP biogenesis and mRNA splicing [15,16].

RNA editing is a biological process through which the nucleotide sequence specified in the genomic template is modified to produce a different one in the transcript, thus contributing to proteomic sequence variation. The two best-characterized classes of RNA editing are Cytydine-to-Uracyl and Adenine-to-Inosine editing (C-to-U and A-to-I, respectively).

Altogether, these RNA modifications represent a new layer of control of genetic information and can affect a variety of molecular processes, thus indicating the regulation of RNA modification as a crucial step in the tight regulation of gene expression, mRNA stability, processing, and translation efficiency [17].

Three types of effector proteins are responsible for the regulation of RNA modifications: (i) writers proteins, also known as RNA-modifying enzymes, which catalyze the transfer of chemical groups on the RNA target; (ii) reader proteins, that are RNA binding proteins that specifically bind the modified nucleotide; (iii) eraser proteins, which remove specific chemical groups from the modified nucleotides, converting them back into unmodified nucleotides [18,19]. In this context, proteins involved in RNA editing are also defined as “editors”. A summary of the better studied and characterized epitranscriptomic determinants is reported in Table 1.

In the present review, we will discuss state-of-the-art on the role of epitranscriptomics in normal and malignant hematopoiesis, focusing on the dysregulation and mutations occurring in RNA modification enzymes.

## 2. Methods for Detection and Mapping of RNA Modifications

The advances in epitranscriptomic knowledge have mainly been triggered by the advent of next-generation sequencing (NGS) and the supporting bioinformatics tools. The first RNA nucleoside modification was identified in 1957 [20] by paper chromatography. Earlier methods mainly relied on the analysis of nucleoside digests of purified RNA by thin-layer cellulose plates or liquid-column chromatography.

The recent, highly sensitive mass spectrometry and high-performance liquid chromatography methods have allowed for the detection of quite low amounts of non-canonical nucleotides in RNA digests [21].

Methodologies currently used to study the distribution and mapping of the most common RNA modifications can be classified in two main strategic approaches.

The first approach regards the identification of methylated nucleosides (m6A, m5C, m1A), which is obtained through immunoprecipitation techniques, and their variants, mainly using antibodies against methylated nucleosides or associated proteins in vitro. In these approaches, RNA is fragmented, and then fragments containing modified nucleoside are enriched before the sequencing. NGS analysis of the immunoprecipitate allows for the identification and mapping of the modification. The resolution of the data can range depending on the specific protocol used from 100–200 nt to base-resolution [22].

The second approach is used to detect modifications other than methylation. Generally, modified nucleosides are chemically modified, and the resulting RNA is retrotranscribed. During reverse transcription, transcriptase activity stops when a chemically modified nucleoside is encountered, yielding a truncated cDNA molecule. Truncated cDNA sequencing allows mapping of the specific modification.

In this context, it is essential to cite the potential contribution that could derive from the most recent direct RNA nanopore sequencing technology [23,24]. Today, the Oxford nanopore sequencing platform is the sole technology that directly sequences RNA molecules. Together with its ability to generate RNA long-read sequencing, which has proved useful for characterization of the isoform, this technology can detect modified nucleosides in the RNA during sequencing [25,26]. In our opinion, this technology will substantially boost a more comprehensive knowledge in the epitranscriptomics field.

## 3. Epitranscriptomics and Hematopoiesis

Hematopoiesis is defined as a regulated process occurring in a small pool of multipotent hematopoietic stem cells (HSCs) that produce differentiated blood cells. After decades of research and exploration, scientists have acquired a basic knowledge of in vivo development and in vitro differentiation of HSCs, but the understanding of the dynamic regulation mechanism of the entire process is not yet perfect, especially regarding epigenetic modification in vertebrate hematopoiesis. The role of stem cell development is little known. Epitranscriptomic mechanisms in hematopoiesis highlight new scenarios that may boost the understanding of hematological diseases. Recent studies have shown that chemical modifications on RNA contribute to gene expression control. m6A modification is the most prevalent chemical mark in eukaryotic mRNAs [27]. RNA metabolism, more specifically RNA stability and translation, is controlled by m6A-dependent regulatory pathways. The methyltransferase-like 3 (METTL3) writer regulates proliferation and differentiation in HSCs. In healthy human HSCs, loss of METTL3 leads to a reduction in m6A levels that results in decreased proliferation and enhances myeloid differentiation. On the other hand, overexpression of wild-type METTL3 increases m6A levels and inhibits cell differentiation, promoting proliferation. The enzymatic activity of METTL3, and therefore m6A, is crucial to maintaining these characteristics, probably through modulation of the PI3K/AKT signaling pathway [28]. Studies of the expression patterns of m6A writers and erasers in different populations of mouse bone marrow (BM) cells and of gene function in human CD34+ HSCs have highlighted that methyltransferase-like 14 (METTL14) also plays a role in inhibiting normal myelopoiesis. METTL14 is highly expressed in HSCs, promoting self-renewal of normal HSCs and leukemia stem/initiating cells (LSCs/LICs), and is downregulated during normal myelopoiesis. METTL14 regulates mRNA stability and the translation of MYB and MYC genes, players in the self-renewal and differentiation of normal HSCs and acute myeloid leukemia (AML) cells. Competitive transplantation assays showed that HSCs lacking METTL14 exhibited reduced repopulation capacity, suggesting a partial inhibition of self-renewal [29]. Moreover, Weng et al. have pointed out the role of Sp1 protein, an inducer of granulocytic/monocytic differentiation, which negatively affects METTL14 transcription [29]. In the hematopoietic scenario, there are many proteins involved as m6A regulators such as RBM15, DNMT2, and Angiogenin (ANG). RBM15 guides the methylation of adenosine residues in both mRNAs and the lncRNA target XIST. The complex RBM15/RBM15B (its paralogue) binds and recruits the Wilms tumor 1-associated protein (WTAP)-METTL3 complex to specific sites. Moreover, RBM15 is differentially expressed during hematopoiesis and may inhibit myeloid differentiation through Notch signaling stimulation in hematopoietic cell lines. The Notch signaling pathway is related to cell fate specification, differentiation, proliferation, and survival. In the study by Ma et al., the knockdown of RBM15 led to more rapid myeloid differentiation, suggesting that Notch is involved in inhibiting the myeloid lineage [30]. DNMT2 has already been reported as a writer for m5C. It seems to play an essential role during hematopoiesis through the methylation of several tRNAs at cytosine 38, which is vital for the fidelity of tRNA. DNMT2-deficient mice failed to achieve endochondral ossification and show a reduction of the HSCs population and defects in their differentiation [31]. ANG, also known as ribonuclease 5 niche-secreted RNase, cleaves the tRNA in half, giving rise to small non-coding RNAs derived from the 5′ end of the tRNAs that accumulate. In contrast, the 3′ tRNA fragments degrade inside the cells. The molecular function of 5' tRNA fragments is to repress cap-dependent translation. Goncalves et al. discovered another role of Ang: it can reduce the proliferative capacity of HSCs while it simultaneously increases the proliferation of myeloid-restricted progenitor (MyePro) cells. Thus, it has been postulated that Ang promotes tRNA production in Lin− Sca-1+ c-Kit+ (LKS) cells (an early form of mouse/murine HSCs), inducing enhanced stemness in vitro and in vivo by reducing the level of global protein synthesis in HSCs. In contrast, Ang stimulates rRNA transcription in MyePro cells, giving rise to a consequent increase in protein synthesis and proliferation [32]. Ang is also involved in another tRNA cleavage mechanism together with the Dicer endoribonuclease producing tRNA-derived fragments (tRFs), which have long been considered pure degradation intermediates of full-length tRNAs. Recent evidence has demonstrated their role in metabolism, immune activity, and stem cell fate commitment. In this context, the Ψ “writer” PUS7 controls protein synthesis and hematopoietic differentiation through the activation of mini 5′ terminal oligoguanines (mTOGs). mTOGs are short-length endogenous 5′ tRNA-derived fragments that modify and activate a new network of tRFs. The dysregulation of this post-transcriptional regulatory circuitry may contribute to human myeloid malignancies (Figure 1) [33]. Chromosome 7 anomalies have been reported in approximately 10% of cases of de novo and up to 50% of therapy-related MDS, and they are correlated with worse prognosis and reduced overall survival, characterized by high rates of transformation to aggressive leukemia. Guzzi et al. reported that chromosome 7 abnormalities were associated with PUS7 loss as a cause of increased MDS. The authors showed that PUS7 depletion leads to a reduction in mTOGs pseudouridylation and a successive general aberrant translation initiation. The result of this process is an increased protein synthesis in HSPCs that promotes stem cell growth and inhibits differentiation [33]. Additional evidence suggests that 45% of tRFs are significantly enriched in extracellular vesicles (EVs) released by T cells. Antibody-stimulated T lymphocytes secrete EVs significantly enriched for 5ʹ-tRFs; the accumulation of 5ʹ-tRFs is detrimental to T cell activation because of the reduced synthesis of co-stimulatory cytokines such as interleukin 2 [34]. Further studies in mice have proven that the RNA-editing enzyme adenosine deaminase acting on RNA-1 (ADAR1) gene ablation causes a significant defect in the final stages of B cell differentiation that leads to a complete absence of newly formed immature and CD23+ mature recirculating B cells in the bone marrow [35]. More generally, genetic ADAR1 deletion induces embryonic lethality in mice by impairing normal hematopoiesis [36]. Instead, ADAR1 deletion in adult mice increases interferon signaling, resulting in HSC exhaustion [37,38]. All these RNA modifications provide a way to regulate transcripts involved in many cellular roles such as switching on cell-differentiation programs.

## 4. Epitranscriptomics in Hematological Ma Lignancies

As reported below, the most relevant RNA chemical modification is the N6-methylation of adenosine. N6-methyladenosine internal changes are abundant in eukaryotic mRNAs and post-transcriptionally regulate the expression of thousands of mRNA transcripts, dynamically and reversibly, during many major normal bioprocesses (self-renewal/differentiation of embryonic stem cells and HSCs, tissue development, circadian rhythm, heat shock/DNA damage response, and sex determination) [39]. Alterations in m6A machinery are strongly associated with the pathogenesis and drug response in solid and hematologic diseases (i.e., AML). While m6A functions and mechanisms in hematopoiesis remain elusive, it is clear that its dysregulation can impair the proper differentiation program.

AML is a heterogeneous and aggressive clonal disease of HSCs and primitive progenitors that blocks their myeloid differentiation, generating self-renewing LSCs. AML is the most carefully investigated hematological malignancy in epitranscriptomics; in particular, deregulation of the m6A modifying system (writers, readers, erasers) can contribute to the development and progression of AML [40]. The writers METTL3 and METTL14 are the critical regulators of differentiation in both normal hematopoiesis and AML pathogenesis [41]. In particular, METTL3 mRNA and protein have been reported to be highly expressed in AML cells suggesting its role in myeloid differentiation in both normal and leukemic hematopoiesis [28,42]. In this context, METTL3 acts on the general transcripts’ methylation status, but also regulates Wtap homeostasis (Figure 1). On the other hand, Wtap is a regulatory subunit of the m6A methylation complex. Wtap is a highly conserved protein that partners with WT1, a switch oncogene, regulating the cell balance between quiescence and proliferation. It plays a crucial role in leukemogenesis and has been correlated with poor prognosis in AML. Wtap levels were not associated with particular cytogenetic abnormalities, but a significant correlation was found between some specific molecular mutations such as NPM1 and FLT3-ITD, and WTAP expression [43]. WTAP is commonly upregulated in many tumors including AML, but its upregulation alone is not enough to promote cell proliferation in the absence of a functional METTL3. Alterations in the expression level of METTL3 protein result in WTAP upregulation and homeostasis [44], suggesting a further oncogenic role of WTAP in supporting METTL3 activity.

METTL14 has been reported to be highly expressed in normal HSCs and AML and downregulated during myeloid differentiation. In particular, METTL14 expression resulted in being elevated in AML cells carrying 11q23 alterations, t(15;17), or t(8;21). Weng et al. postulated that METTL14 promotes oncogenesis by regulating some mRNA targets such as MYB and MYC through m6A modification (Figure 1). On the other hand, METTL14 is negatively regulated by Sp1, which is normally expressed during myeloid and B-lymphoid cell development [29]. Together with m6A writers, eraser and readers are also reported to be involved in leukemogenesis.

Li et al. showed that the fat mass and obesity-associated (FTO) m6A eraser protein results in being commonly highly expressed in AMLs with 11q23/MLL rearrangements, t(15;17)/PML-RARA, FLT3-ITD, and/or NPM1 mutations and that these oncogenic proteins directly induce FTO gene upregulation [45]. In turn, FTO upregulation enhances the expression of ASB2 and RARA by reducing m6A levels at untranslated regions (UTRs), thus inhibiting all-trans-retinoic acid (ATRA)-induced AML cell differentiation and promoting leukemogenesis (Figure 1). Mechanistically, m6A modifications are associated with a stable mRNA reduction, but the authors do not exclude other auxiliary “reading processes” [45]. The possible relationship between FTO expression, overweight/obesity, and the risk of various types of cancers including hematopoietic malignancies such as myeloma, lymphoma, and leukemia is interesting. Some authors have reported increased FTO expression levels associated with single nucleotide polymorphism (SNP) rs9939609 and an increased risk of obesity and diabetes [46]. This observation could match with observations from studies that show an association between the risk of hematological malignancies and diabetes and/or overweight/obesity [47,48]. On the other hand, other authors have not found rs9939609 SNP to be associated with increased cancer risk [49]. It would be useful to investigate this point further and elucidate whether rs9939609 SNP can be considered a characteristic predisposing to cancer. Apart from FTO, ALKBH5 is another m6A eraser that has been shown to be pivotal for tumorigenesis and cancer stem cell self-renewal in AML (Figure 1). ALKBH5 has been found to be overexpressed in human AML, and its deletion is rare in AML and is associated with poor prognosis. According to these studies, ALKBH5 promotes leukemogenesis and LSC/LIC self-renewal through m6A-dependent mechanisms [37]. The m6A reader YTHDF2 has been found overexpressed in many AMLs and seems to be required for disease initiation and propagation. This reader would, therefore, act by decreasing the half-life of different m6A transcripts that contribute to the overall integrity of LSC. Additionally, YTHDF2 is not essential for normal HSC function because its deficiency enhances HSC activity by regulating the stability of multiple mRNAs that are critical for HSC self-renewal while inhibiting LSCs expansion [50]. Apart from AML, which is the most closely epitranscriptomically studied hematological malignancy, other studies have focused on other malignancies, but overall, they are still in their infancy. In mantle cell lymphoma (MCL), the expression levels of one-half of the m6A regulators may be used to predict patient survival. A low m6A-index was reported, associated with poor patient survival and lower mRNA levels from the total transcriptome in MCL [51]. Moreover, the m6A modification plays a functional role in regulating the stability of viral latent and lytic transcripts in a wide range of human cancers induced by Epstein-Barr virus (EBV). METTL14 was reported to be significantly overexpressed at the transcript and protein levels in EBV-positive tumors. Ebna3c viral onco-protein activates the transcription of METTL14 during the EBV-mediated tumorigenesis [52]. It is well known that EBV establishes latent infection in human B cells and can transform B lymphocytes through EBV transcription factors action including EBNA3 proteins [53]. Thus, METTL14 may represent a new therapeutic target for the treatment of EBV-associated cancers. Other epitranscriptomics studies were conducted on Burkitt’s lymphoma (BL). BL acquires consistent point mutations in a conserved domain of MYC, MYC Box I. It has been reported that NOL5A/NOP56, a ribosomal RNA methylation gene, results in being hyperactivated in MYC mutated BL patients. Furthermore, NOL5A enhances MYCWT-induced cell transformation [54]. Small nucleolar RNAs (snoRNAs) and small Cajal body-specific RNAs (scaRNAs) are non-coding RNAs involved in the regulation of various types of modifications borne by ribosomal RNAs and spliceosomal RNA, respectively, generally located in the introns of host genes. Alterations of sno/scaRNA expression may play a role in cancerogenesis and hematological diseases. A global sno/scaRNA downregulation was found in multiple myeloma [55]. In chronic lymphocytic leukemia (CLL), the pattern of sno/scaRNAs expression is unrelated to the major biological (ZAP-70 and CD38), molecular (immunoglobulin heavy chain gene mutation), and cytogenetic markers. In contrast, SNORA74A and SNORD116-18 expression appear to be significantly downregulated in two different CLL groups with poor prognosis [56]. Furthermore, the snoRNAs resulted in being deregulated in the tumor cells of some subtypes of peripheral T-cell lymphoma (PTCL), a rare and heterogeneous type of non-Hodgkin lymphoma, associated with a poor clinical outcome. Mainly, snoRNAs are globally downregulated in PTCL-not otherwise specified (PTCL-NOS), and angioimmunoblastic T-cell lymphoma (AITL), but a specific snoRNA, HBII-239, was significantly over-expressed in cases of AITL and PTCL-NOS with favorable outcomes [57]. Thus, sno/scaRNAs deregulation may contribute to predict the clinical outcome, offering new tools for patient care and follow-up. Finally, in chronic myeloid leukemia (CML), it is well known that patients respond well to tyrosine kinase inhibitors (TKIs) of the BCR-ABL oncoprotein, without eradicating the disease due to rare TKIs insensitive LICs. LICs in CML can arise from myeloid progenitor cells, a population that depends on the ADAR1 (Figure 1). A study in a conditional ADAR1 knockout mouse model showed that the ADAR1 depletion normalized the peripheral white blood count by eliminating leukemic BCR-ABL transformed cells [58]. Zipeto et al. showed in a mouse blast crisis (BC) model that let-7 microRNA editing is a pivotal process in leukemia stem cell self-renewal. The impairment of let-7 is dependent on JAK2 and BCR-ABL-mediated activation of ADAR1 [59]. 

## 5. A Therapeutic Opportunity

Although epitrascriptomics knowledge is in its infancy, the scientific community and the pharmaceutical industry are aware of its therapeutic relevance, particularly in developing an inhibitor of METTL3, which has shown reduced splenomegaly and number of circulating monocytes in a mouse model of AML as well as slowed patient-derived xenograft growth. Phase I trials are expected for 2021 [60].

The more investigated epitranscriptomic effector in terms of therapeutic agents is FTO. Rhein was the first inhibitor of FTO, showing, for the first time, the chance to modify the mRNA methylation in cells [33]. Subsequently, more selective and potent inhibitors were identified. In this regard, Fb23 and its derivative Fb23-2 directly bind to FTO and selectively inhibit FTO’s m6A demethylase activity. Fb23-2, in particular, has shown to suppress progression and promote apoptosis of primary blast AML cells in vitro and xenotransplanted mice [61,62]. Moreover, (R)-2-hydroxyglutarate (R-2HG) produced by IDH mutants is another FTO signaling inhibitor and has been shown to suppress AML progression.

Finally, some authors proposed ADAR1 as a new therapeutic target together with Bcr-Abl and JAK22 in the treatment of TKI-resistant CML and BC CML [59]. Recently, it has been announced that an ADAR1 inhibitor phase I trial will start in 2022.

Little is yet known about the epitranscriptomic mechanisms underlying hematological malignancies, but their understanding could offer significant therapeutic opportunities, thus raising the need for further investigations.

## 6. An Evolutionary Vision

In 1973, Theodosius Dobzhansky published the essay, “Nothing in biology makes sense except in the light of evolution” [63] to highlight that evolution is a crucial unifying principle in biology. Epitranscriptomic events and its determinants (writers, readers, and eraser proteins) can control a wide range of cell activities by modulating the fate of the gene transcripts. What are the evolutionary bases of the epitranscriptomics? What is the need for an additional level of control of gene expression? As already reported for epigenomics, many epitranscriptomic processes could have evolved as part of a host defense mechanism against non-self pathogens such as viruses and parasitic DNA [64]. The ADAR1 protein is the best-characterized example of the use of RNA modifications to discriminate self/non-self RNA. Adar RNA-editing enzymes convert adenosine (A) to inosine (I) [65]. ADAR1 activity prevents aberrant activation of the innate immune system due to the presence of Alu elements in the 3'UTR of some transcripts. Indeed, Alu elements can form RNA duplexes that can activate the immune response. The ADAR1 protein activity establishes innate immune tolerance for host dsRNA [66]. In other words, innate immune sensors can be considered as readers of epitranscriptome RNA modifications, and the use of host RNA modification to distinguish between host and parasite nucleic acids is reminiscent of the restriction and modification system that occurs in bacterial DNA [66]. Other authors have speculated that ADAR1, commonly upregulated in many tumors including some hematological malignancies could promote cancer progression through the regulation of type I interferon (IFN) and its induced gene signature. Thus, an increase of ADAR1 expression may hide the cancer cells, allowing them to escape recognition by the melanoma differentiation-associated protein 5 (MDA5) and protein kinase R (PKR) pathways, as they will no longer accept these as substrates [67]. Moreover, other authors have recently shown that loss of function of ADAR1 in tumor cells profoundly sensitizes tumors to immunotherapy and overcomes resistance to checkpoint blockade [68]. In the study by Steinman et al. discussed below, both immature and total leukemic cells were more vulnerable to death after ADAR1 deletion, whereas their ADAR1-floxed counterparts were not transformed using BCR-ABL. The regression of established CML in the ADAR1 deleted mouse model has been associated with the loss of hyper editing of non-coding RNA activating interferon signaling and/or apoptosis [58,69]. In this context, some authors have asserted that since RNA m6A occurs in the nucleus, it could be used as a marker to discriminate endogenous RNA from exogenous RNA [70,71]. Rig-I-like receptors Rig-I and Mda-5 and Toll-like receptors (TLRs) recognize foreign RNAs through specific features and activate the response that leads to cytokine induction. RNA modifications including m6A m5C, m5U, or pseudouridine are presumed to play a role in this system. Indeed, dendritic cells (DCs) exposed to such modified RNAs express significantly fewer cytokines and activation markers than those treated with unmodified RNA [72]. Moreover, it has been reported that viruses can adopt strategies to evade immune recognition, for example, for this purpose, the human metapneumovirus uses m6A modifications in its genomic RNAs and when specific m6A is mutated in vitro, a higher IFN production is induced after infection [73].

In AML and HSCs, the METTL3 role has been reported to sustain pluripotency by modulating mRNA stability. Its upregulation in AML cells would induce m6A of mRNAs such as MYC, BCL-2, and Pten, which are necessary for regulating apoptosis and differentiation, leading to their efficient stability and translation (Figure 1) [28]. Similarly, METTL14 in normal HSPCs and AML would regulate mRNA stability and translation of the essential targets MYB and MYC [29]. Moreover, Huang et al. reported that insulin-like growth factor 2 binding proteins (IGF2BPs), by acting as m6A readers, should promote the stability and translation of target transcripts at the post-transcriptional level [29]. Finally, FTO has been reported to have a negative effect on two crucial tumor suppressors, ASB2 and RARA, in a m6A-dependent manner that could promote mRNA stability [45]. Thus, according to these observations, the dysregulation of epitranscriptomic determinants could promote immune tolerance or an anti-apoptotic effect for tumor cells including leukemia [68]. The incidence of hematological malignancies, together with most cancers, increases with age. In other words, they are considered as post-reproductive age diseases, which means that this phenotypic trait is subject to selective pressure at different levels. Thus, epitranscriptomic events could act at both the gene regulation level and immune surveillance level. Dysregulation of the epitranscriptomic determinants could act as passenger events and could confer selective advantages, promoting the progression of leukemogenesis and conferring immunological escape features. Thus, a more extensive knowledge of the epitranscriptomic mechanisms will be essential to better understand the processes that lead to the establishment of malignancies, thus contributing to improve immunotherapeutic and targeted treatments.

## Figures and Tables

**Figure 1 ijms-21-06578-f001:**
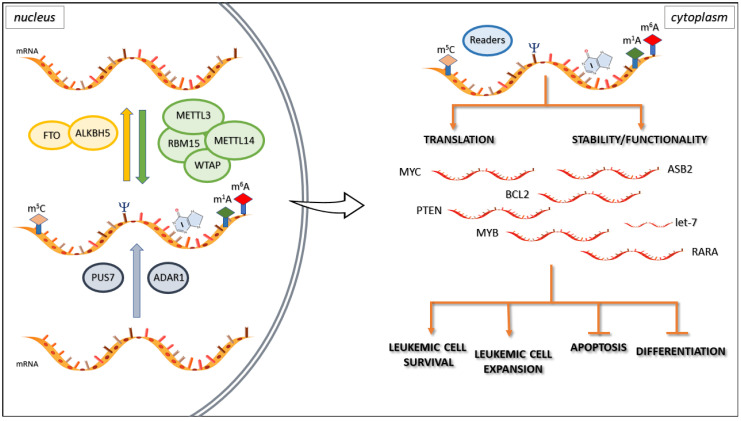
Schematic depiction of the principal components of epitranscriptomic mechanisms and their involvement in malignant hematopoiesis. Writers and erasers of main reversible methylations are represented in green and yellow, respectively. ADAR1 and PUS7 are the main enzymes involved in irreversible modifications (blue): ADAR1 converts adenosine to inosine, and PUS7 promotes pseudouridylation. Dysregulated modification affecting mRNA translation or RNA stability and/or more general functionality of some key RNA molecules promotes the leukemic phenotype.

**Table 1 ijms-21-06578-t001:** Principal epitranscriptomics determinants.

Modification	Writers	Readers	Eraser
m6A	METTL3		
	METTL14		
	WTAP	YTH domain proteins	FTO
	RBM15	IGF2BP family	ALKBH5
	RBM15B	HNRNPA2B1	
	KIAA1429		
m5C	NSUN1/2/3/4/5	ALYREF	Still not identified
	DNMT2	YBX1	
m1A	TRMT6/10C/61A	YTHDF1/3	ALKBH1/H3
		YTHDC1	
PseudoUridine	PUS genes	Not identified	Not identified possible irreversible modification
	DKC1		
A-to-I Editing	ADAR genes	Not identified	Not identified possible irreversible modification
	ADAT genes		
C-to-U Editing	AID/APOBEC gene family	Not identified	Not identified possible irreversible modification

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
