# Peer review of "Epitranscriptomics in Normal and Malignant Hematopoiesis"

_ijms, 2020, doi:10.3390/ijms21186578_

Round 1

Reviewer 1 Report

In this review, the author summarizes early and recent findings in terms of epitranscriptomics in normal and malignant hematopoiesis. This review is providing a wealth of information. This reviewer has no major criticism of this comprehensive work.

Here is one minor comment on the style of this paper.

The last paragraph of “4. Epitranscriptomics in hematological malignancies”

The first paragraph of “5. An evolutionary vision”

These two paragraphs contain various different topics which are not directly relevant. This reviewer recommends the long paragraphs be segmented according to specific topics.

Author Response

The section “4. Epitranscriptomics in hematological malignancies” has been modified as suggested (line 304-309 page 8 and the new section “5. A therapeutic opportunity” on page 8). In our opinion, the addition of further paragraphs, thus fragmenting “5. An evolutionary vision” section it would not improve the structure of the manuscript.

Reviewer 2 Report

This is a well written and comprehensive review detailing an emerging field in hematopoietic regulation. It will be a nice addition to the literature on epitranscriptomics.

1. It would be interesting (but not required) to summarize current understanding of the role of epitranscriptomics in the context of hematopoietic regeneration, particularly in the context of post-transplantation. This was recently reviewed and could be included (doi: 10.1097/MOH.0000000000000585). 

2. It should be noted that a similar review was recently published in Blood Cancer Discovery (DOI: 10.1158/2643-3249.BCD-20-0032). This certainly does not preclude this paper from also being accepted, but the authors may want to consider also including some details on the role of epitranscriptomics in MDS (DOI: 10.1016/j.cell.2018.03.008) in addition to AML. 

3. It would be interesting to summarize the implications that emerging understanding will have on therapeutic development. This could be a stand alone section that concludes the review.

Author Response

…It would be interesting (but not required) to summarize current understanding of the role of epitranscriptomics in the context of hematopoietic regeneration, particularly in the context of post-transplantation. This was recently reviewed and could be included (doi: 10.1097/MOH.0000000000000585)…” 

The text has been modified, as suggested (line 125-127 on page 3).

“…It should be noted that a similar review was recently published in Blood Cancer Discovery (DOI: 10.1158/2643-3249.BCD-20-0032). This certainly does not preclude this paper from also being accepted, but the authors may want to consider also including some details on the role of epitranscriptomics in MDS (DOI: 10.1016/j.cell.2018.03.008) in addition to AML…” 

The text has been modified, as suggested (line 156-163 on page 4).

“…It would be interesting to summarize the implications that emerging understanding will have on therapeutic development. This could be a stand alone section that concludes the review…”

We have created a new section on therapeutic development, as suggested.

Reviewer 3 Report

This is a clearly written and comprehensive review on a novel, revelant topic. I have only one minor comment:

- abstract, last sentence, line 21: "This review focuses on the role of epitranscriptomics events that are known implicated.." I guess a "to be" is missing between "known" and "implicated"

Author Response

“…abstract, last sentence, line 21: "This review focuses on the role of epitranscriptomics events that are known implicated.." I guess a "to be" is missing between "known" and "implicated"”

The text has been corrected, as suggested